# Ion Separation Together with Water Purification via a New Type of Nanotube: A Molecular Dynamics Study

**DOI:** 10.3390/ijms24076677

**Published:** 2023-04-03

**Authors:** Jiao Shi, Xin Zhou, Pan Jia, Kun Cai

**Affiliations:** 1Key Laboratory of Agricultural Soil and Water Engineering in Arid and Semiarid Areas, Ministry of Education, Northwest A&F University, Yangling 712100, China; 2College of Water Resources and Architectural Engineering, Northwest A&F University, Yangling 712100, China; 3School of Science, Harbin Institute of Technology, Shenzhen 518055, China

**Keywords:** capacitive deionization, carbon nanotube, concentric twin tube, nanochannel, molecular dynamics

## Abstract

We propose a CNT-based concentric twin tube (CTT) as nanochannels for both water purification and ion separation at the nanoscale. In the model, a source reservoir dealing with the solution connects three containers via the CTT that has three subchannels for mass transfer. Before entering the three subchannels, the solution in the separating zone will form three layers (the aqua cations, water, and the aqua anions, respectively) by applying a charged capacitor with the two electrodes parallel to the flow direction of the solution. Under an electric field with moderate intensity, the three subchannels in the CTT have stable configurations for mass transfer. Since the water and the two types of aqua ions are collected by three different containers, the present model can realize both ion separation and water purification. The mass transfer in the subchannels will be sped up by an external pressure exerted on the solution in the source reservoir. The physical properties of the model, e.g., water purification speed, are analyzed with respect to the effects of the electric field, the size of CTT, and the concentration of solute, such as NaCl.

## 1. Introduction

On our planet, only 2.5% of water is fresh water. The drinkable water is about 3% of fresh water. Nowadays, 40% of the world’s population faces water scarcity and/or serious water pollution. The situation will be getting worse in the coming years. To overcome the challenge, water purification technology is vital and some techniques have been successfully developed in the last decades. Distillation and membranes are the major techniques for water purification. At present, about 60% of the drinking water obtained by desalination is produced in distillation plants. As large heat energy is required in distillation, the worldwide application of the method [1,2] is confined. When using membranes (for osmosis [3,4] or nanofiltration [5,6]) for large-scale desalination of water, external pressure is required to overcome the natural osmotic pressure which pushes the water through the membrane. Since the nanofiltration technique can only remove divalent ions, it is not suitable for seawater desalination while reverse osmosis has a high cost on membrane and energy consumption. Hence, the electrodialysis method was proposed. In the method, a capacitor was introduced for separating ions [7,8], in which ion flux was controlled by two different exchange membranes that face the two electrodes. When the two electrodes were made from the carbon materials that have a high surface area, the method was called capacitive deionization (CD), in which the ions were first separated from the solution and then released as a waste stream [9,10]. When the membranes were inserted on the two electrodes, the technology is named as membrane capacitive deionization (MCD) [11,12,13]. The two models have a small difference, and MCD was better for desalination at a lower energy consumption. Moreover, the ion separation can be used to produce a battery. For developing a battery with higher output power during water purification, other methods, like microbial desalination cells [14,15,16,17,18], were developed.

It is known that in water purification by a membrane technique, the major element of cost is the membrane. Hence, developing new filtration techniques on the nanoscale has attracted much attention in recent years. Having excellent permeability to water and gas [19], the nanochannels in membranes are widely applied in various fields, e.g., seawater purification [20,21], nanofluid devices with high throughput [22,23], sensors [[24],[25]，[26]], and protective clothing [27,28]. A consensus on mass transfer is that the structure and dynamic behavior of fluid via nanochannels are different from those on the macroscale. For example, high-speed water transfer was observed in the nanochannels made from such materials as slim carbon nanotubes (CNTs), pumice, and ion channels in biofilms [29,30,31]. In this study, we adopt CNTs to build a new type of nanochannel for water purification owing to CNTs’ excellent mechanical properties, thermal and chemical stabilities, and extremely low friction during relative motion [32,33,34] or mass transfer [19]. For example, the ratio of desalination of the CNTs is higher than 95% when their diameters are between 1.0 and 1.1 nm [35]. However, the ratio of desalination will drop sharply when increasing the diameters of CNTs. The ratio can be lower when improving the solute concentration, which produces higher resistance on the motion of the aqua ions and water molecules [36] and fewer ions attaching to the surface of CNTs [37].

The geometry of CNTs also affects mass transfer. For example, the wall thickness of a CNT influences the transfer of aqua ions. It is caused by the dehydration process of the ions passing through the nanochannels [38]. The chirality of CNTs has a unique electron configuration on the inner surface of CNTs [39] which governs the resistance at the entrance of the nanochannel and the interfacial friction. In general, the chirality effect becomes weaker when the tube diameter is higher [40]. The CNTs with funnel-shaped openings have a ratchet effect [41] that assists the nanoparticles entering the tubes. The fluid structure and the type of ions can be adjusted by decorating the CNTs with specific functional groups [42,43,44]. However, the aquation of the ions in the solution weakens the permeability of CNTs. If improving the permeability by adopting the CNTs with larger diameters, the capability of ion separation will be weakened. The competition between permeability and separation quality can be avoided by the introduction of an external electric field to separate the ions before entering the nanochannels.

In the above studies, the nanochannels have no subchannels that are separated for ion separation. When the external electric field is removed, the aqua cations, water, and the aqua anions in the single channel mix immediately. It implies that the transfer of cations fails through a nanochannel in a nanoscale environment, e.g., a biological cell. This study proposes a new ion separation model featured with nanochannels made from concentric twin tubes (CTT) and an external electric field for ion separation. The concentric twin tube can be fabricated by inserting a CNT into another identical CNT (twin tube). A relaxed CTT has three subchannels [45,46], which are suitable for the transfer of positive and negative aqua ions and water, respectively. The molecular dynamics simulation approach is adopted to look into the physical properties of the new nanochannel model. The models and methodology are introduced in Section 2. The numerical discussion is given in Section 3. Some conclusions are drawn in Section 4.

## 2. Model and Methodology

### 2.1. Model of an Ion Separation System

Figure 1 illustrates an ion separation system. The CTT channel contains three subchannels in the overlap area, denoted as A, B, and C, with a length of 30 Å. For a CTT made from armchair nanotubes with the chiral index of (m, m), the CTT is named as “**A-m**”. For example, the A-30 CTT is made from two (30, 30) CNTs. The solution in Ω_1_ (the source reservoir) will flow into Ω_2_ when the valve between the two zones is open and piston starts moving rightward. Note that the valve is not required in a real nanosystem. Herein, the function of valve is to confine the solution in Ω_2_ before the solution reaching a reasonable state. The chloridions (Cl^−^) and sodions (Na^+^) in Ω_2_ will be separated by the electric field generated with the capacitor. The positive ions will move closer to the Ω_3C_ subchannel, while the negative ions to the Ω_3A_ subchannel. Under the pressure from the piston, the nanoflow moves through the subchannels and then enters the containers. For example, the ions in Ω_3A_ enter Ω_5A_ via Ω_4A_. Similarly, the ions in Ω_3C_ enter Ω_5C_ via Ω_4C_. Theoretically, the concentration (*C*) of the ions in Ω_3B_ is much lower than that in Ω_1_. Therefore, both the ion separation and the purification of water can be realized through the CTT-based nanochannel. If dielectric materials are adopted as a frame to confine the solution, the separated ions can generate electricity. This is the third function of the present model (the three-dimensional model is attached as a Appendix A).

For simplicity, only the CTT inner tube in the system is deformable. Along the direction of the electric field, a non-periodic and fixed boundary is applied to the unit cell. The other two dimensions are applied with periodic boundaries with the gap more than 15 Å each side. The intensity of the electric field, referred to as *E*_EF_, is closely related to the force on the ions. For instance, a stronger electrostatic force may result in the stacking of the ions in Ω_2_, and further, the subchannels will be stuck by the ion stack. If the force is too weak, the positive and negative ions cannot be separated efficiently before entering the subchannels. Hence, the effect of the electric field intensity on the ion separation needs to be elaborated. The concentration of NaCl (*C*_NaCl_) in solution in the source reservoir, as an essential factor, will be discussed in detail the water molecules and ions listed in Table 1.

### 2.2. Methodology

The molecular dynamics simulation is adopted for revealing the motions of the ions in the subchannels. The simulations are conducted on the open-source code LAMMPS (Large-scale Atomic/Molecular Massively Parallel Simulator) [47]. The simple point charge-extended SPC/E water model [48] is for describing the behavior of water molecules [49,50] with the SHAKE algorithm [51] to keep water molecules as rigid bodies. The Joung-Cheatham force field [52] is for describing interaction among the monovalent ions, i.e., Na^+^ and Cl^−^. The interaction among the carbon and carbon or hydrogen atoms is evaluated by the AIREBO potential [53]. The Lorentz-Berthelot rule [54] is used to calculate the cross interaction between two nonbonding atoms with the interaction potential parameters listed in Table 2. The cut off distance of nonbonding interaction is set to be 12 Å and the particle-particle particle-mesh (PPPM) method is used to calculate the long-range electrostatic interaction. In a simulation, the Nosé-Hoover thermostat [55,56] is applied to maintain the system temperature. The time step is set to be 0.001 ps and the result is output every 1 ps. In each simulation, the following steps are required, i.e.,

Step 1: Build the initial geometry model of the system with a fully relaxed CTT channel.Step 2: Reshape the positions of atoms in the system by minimizing the potential energy of the system.Step 3: Relax the system (**Stage I**) at 300 K by fixing the piston and confining the solution in the source reservoir (Ω_1_) for 300 ps. After this relaxation, the solution reaches a more reasonable state.Step 4: Open the valve to let the solution enters Ω_2_ and keep relaxing the whole system for 500 ps. The system reaches a relatively stable immersion state (**Stage II**).Step 5: In **Stage III**, push the graphene piston at a constant speed (*v*), e.g., 0.01 Å/ps in this study, to force the solution entering CTT channel for separation.Step 6: Record the essential data for post processing.

For reducing the computational cost, the interaction between the hydrogen atoms in water and the carbon atoms on the graphene/CTT-based frame is not considered. The electric conductivity of the carbon components is not considered in simulation, which can be easily realized by replace the boundaries of the carbon components with dielectrical materials, e.g., hexagonal BN.

### 2.3. Physical Quantities of the System in Separation

For tracing the state of the system, herein, some key quantities are calculated during simulation. First, the **variation of potential energy** (Δ*P*) is calculated to describe the state of the whole system. It reads
(1)$$\Delta P(t)=P(t)-P(t_{0})$$
where *P*(*t*) and *P*(*t*_0_) are the current and initial potential energy of the system, respectively.

In the flow of solution, the contact state between the carbon component and the solution can be evaluated by their **interaction energy** (*E*_C-Sol_), i.e.,
(2)$$E_{C-\mathrm{Sol}}=P_{\mathrm{Sol}}+P_{C}-P_{\mathrm{System}}$$
where *P*_System_, *P*_Sol_ and *P*_C_ are the potential energy of the overall system, the isolated solution, and the isolated carbon component, respectively.

The **mean square displacement** (MSD) illustrates the mobility of water molecules and ions:(3)$$\mathrm{MSD}(t)=\frac{1}{N}\langle \sum_{i=0}^{N}{\left|r_{i}(t)-r_{i}(t_{0})\right|}^{2}\rangle$$
where *N* is the number of particles, *r_i_* denotes the *i*th particle position in the flow direction, and angular brackets means an ensemble average.

The **discharge of solution** through the CTT means the total number of solution particles (H_2_O, Na^+^, and Cl^−^) through the three CTT subchannels at time *t*, i.e.,
(4)$$\left\{ \begin{array}{l} d_{\mathrm{Sol}}^{A}(t)=N_{\mathrm{Sol}}^{3A}(t)+N_{\mathrm{Sol}}^{4A}(t)+N_{\mathrm{Sol}}^{5A}(t),\\ d_{\mathrm{Sol}}^{B}(t)=N_{\mathrm{Sol}}^{3B}(t)+N_{\mathrm{Sol}}^{4B}(t),\\ d_{\mathrm{Sol}}^{C}(t)=N_{\mathrm{Sol}}^{3C}(t)+N_{\mathrm{Sol}}^{4C}(t)+N_{\mathrm{Sol}}^{5C}(t), \end{array}\right.$$
where superscripts represent the subchannels or their connected containers Figure 1.

The **water purification rate** is the ratio of the number of water molecules in Ω_4B_ to the total number of water molecules in the three containers, i.e.,
(5)$$R_{\mathrm{Puri}}(t)=\frac{N_{H_{2}O}^{4B}(t)}{N_{H_{2}O}^{4B}(t)+N_{H_{2}O}^{5A}(t)+N_{H_{2}O}^{5C}(t)}$$

The number of hydrogen-bonds (**H-bond**s) (*N*_H-bonds_) among water molecules are calculated, as well. The rule of H-bond between two water molecules *i* and *j* is that the O*_i_*…O*_j_* distance is not greater than 3.5Å and the angle of O*_i_*-H*_i_*…O*_j_* not greater than 30° [57]. The structure and flow characteristics of water cluster highly depend on the topology of the H-bonds in water.

After separation, the electric charge (*Q*) in the three containers (Ω_4B_, Ω_5A_ and Ω_5C_) can be described by the difference of the numbers of sodions and chloridions in each reservoir, i.e.,
(6)$$Q(t)=N_{\mathrm{Cl}-}-N_{\mathrm{Na}+}$$

## 3. Results and Discussion

### 3.1. Effect of Electric Field on Ion Separation

Figure 2a shows four curves of the VPE of the system with respect to different intensity of the electric field transversely applied on zone Ω_2_. It demonstrates that the system deforms slightly when there is no external electric field (*E*_EF_ = 0). This can be verfied by comparing the snapshots at 0.02 and 0.5 ns in Figure 2b. Meanwhile, the ions are not separated before entering the three containers. When *E*_EF_ = 6.9 V/Å, after only 20 ps (i.e., 0.02 ns) Ω_2_ is filled with solution. Some water molecules and ions enter the subchannels and the containers. Hence, the VPE increases rapidly from zero to 0.2 keV. After about 100 ps, the curve of VPE changes slight with time. At 0.5 ns, the three containers have lots of particles entereing from the three subchannels. Meanwhile, the inner tube of the CTT deforms slightly.

When double or treble intensity of the electric field (13.8 or 20.7 V/Å) is applied, the VPE curve jumps sharply in the first 20 ps. It means that the system state has a great change during the short period. By observing the snapshots, we find that the CTT is filled with the solution. The inner tube has a large deformation (Appendix A). Meanwhile, the ions in the source reservoir (Ω_1_) attach to its upper and lower boundaries due to the strong electrostatic forces by the two electrodes. At 0.5 ns, the deformed inner tube almost loses the ability to tranfer the solution (Figure 2b). Hence, an electric field with high intensity is not recommended. In this study, *E*_EF_ = 6.9 V/Å is adopted in the following simulations.

Only when *E*_EF_ = 0 or 6.9 V/Å is applied, the CTT in system constitutes a stable inner tube for mass transfer. We calculate the numbers of water and ions in the three containers as shown in Figure 3a, and find that, as *E*_EF_ = 0, the number difference between the cations and anions (the black and gray curves) is small in all the three containers, which demonstrates that the two types of ions are not separated before entering the separating zone (Ω_3A_, Ω_3B_, and Ω_3C_ in Figure 1). The electric charges of the three containers (Figure 3b) are slightly different from zero. The difference of discharges of solution via the three subchannels (*d*_Sol_ in Figure 3c) is due to their different effective cross sections for mass transfer. Hence, either ion separation or water purification cannot be realized without an external electric field.

When *E*_EF_= 6.9 V/Å, the ions are efficiently separated. For example, Ω_5A_ contains 47 chloridions and three sodions. Hence, the electric charge of the zone is 44 e (the red curve in *Q*-5A, Figure 3b). Simultaneously, Ω_5C_ only has 42 sodions. That suggests the ion separation is successful by the present model. Ω_4B_ contains only one sodion but 2000 water molecules. Thus, the salinity of the solution in Ω_4B_ is 0.162 g/100 mL after neutralization with the same amount of chloridions. Thus, the water in Ω_4B_ is drinkable.

### 3.2. Mass Transfer through the A-30 Channel

For deeply understanding the present model, we investigate the characteristics of motion of the ions and water in the subchannels. VPE of a system (Figure 4a) indicates the change of atom distribution in the system. Due to the existence of the electric field, the value of VPE of either the ions or water increases with time in the first 50 ps in Stage II. Then, the VPE changes slightly. In Stage III, the solution is pushed by the piston, VPE of water decreases first and then increases. Because more and more water enters the three containers, the VPE of water increases. The decrease of the VPE is cause by the water cluster in Ω_1_ being under a growing pressure from the piston. The MSD values of water and ions in Stage II (Figure 4b) increase faster, which demonstrate that the mass flux becomes stronger when piston starts moving. The interaction energy between solution and the carbon components (*E*_C-Sol_ in Figure 4c) decreases monotonously before reaching 5 ns. The sharply increased interaction energy is caused by the particles remained in Ω_1_ which is under a strong compression by the piston.

In Figure 4d, the numbers of water molecules in the three containers (Ω_5A_, Ω_4B_, and Ω_5C_) at the end of Stage II are much lower than those at the end of Stage III. It means that the external pressure on the source reservoir (Ω_1_) is significant for mass transfer via the three subchannels (Ω_3A_, Ω_3B_, and Ω_3C_). In particular, the number of water molecules in Ω_4B_ increases slower than those in Ω_5A_ and Ω_5C_ after 3 ns. It almost keeps unchanged after 3.7 ns because the entrance of Ω_3B_ shrinks and is unsuitable for water transfer (Appendix A). The snapshots in Figure 5b demonstrate the deformed entrance of Ω_3B_ (Figure 5c). According to the numbers of water molecules between 2.5 and 3 ns, the water purification speed of Ω_4B_ reaches 770/ns. The effective area of water transfer of the model equals 59 Å × 106.2 Å(=6265.8 Å^2^). The flux of water purification of the model is 770 mol/(6.02 × 10^23^)/(6265.8 Å^2^)/ns, i.e., 20.41 kmol·m^−2^·s^−1^. It is about 367.44 kg·m^−2^·s^−1^, which is much higher than that of any of the present membranes.

The numbers of the ions in the three containers are listed in Figure 4e, indicating that most of the chloridions enter Ω_5A_, most of the sodions in Ω_5C_, and few ions in Ω_4B_ (e.g., at 4850 ps in Figure 5b). The external pressure applied on the source reservoir gathers the speed of ion separation concluded from the slopes of the curves in Stage II and Stage III. Similarly, the average speed of ion separation by the present model reach 8~9/ns (Figure 5d). During ion separation, the aqua ions (Figure 5d) move fast in the two subchannels Ω_3A_ and Ω_3C_. Hence, the H-bond structure in the water cluster surrounding an ion has a poor stability. For example, the cluster at the left end of Ω_3A/3C_ disperses before leaving the right end of Ω_4A/4C_ (Figure 6).

By observing the ion separation (Figure 5a,b), we also find some interesting phenomena. For example, at the start of relaxation (Stage II), the aqua ions in the separating zone (Ω_2_ and Ω_3_) are attracted by the electrodes, and a bubble is generated at the center part of Ω_2_ (Appendix A). It is a cavitation phenomenon. The solution in Ω_1_ flows into Ω_2_ by attaching the two edges parallel to the electrodes. The bubble extends to Ω_1_ at the first 50 ps. The bubble does not disappear even after the piston starts to move. According to the snapshots in Figure 5b, the bubble disappears between 1300 ps and 1600 ps as more and more solution entering Ω_2_ when the piston keeps moving. Another phenomenon is on the aqueous process of the ions. At 500 ps, the water molecules in Ω_5A_ and Ω_5C_ distribute uniformly. It is because the two containers have a few ions and the distribution of water molecules is not influenced by the electric field. However, at 800 ps, the two zones have more water molecules that form more H-bonds. Most of the water molecules attach the wall that has the exit of Ω_4B_ or Ω_4C_ (Figure 5d) due to the generation of more H-bonds (Figure 4f). As more water molecules entering the two containers with the ions, they gradually form a bubble in each container (3700 and 4550 ps in Figure 5b). Uniformly distribution of the ions in water indicates the aqueous process.

### 3.3. Size Effect of CTT on Ion Separation

When different CTT tubes are adopted as the nanochannel for mass transfer, they have different behavior. Figure 7a indicates that the electric charges of the three containers slightly depend on the tube size when the piston pushes the solution in Ω_1_ at the same speed (Table 3). We also find that the slimmer tubes may have a relatively higher value of electric charge in Ω_5A_ by chloridions and Ω_5C_ by sodions. Since the electric charge in each container illustrates the number difference between the positive and negative ions, it just demonstrates the potential power of the system as a battery. The ion separation effect needs to be explained by the numbers of the positive and negative ions in each container. Figure 7b shows that Ω_5A_ has more chloridions than sodions, while Ω_5C_ has more sodions. The two containers have more ions when a slimmer CTT acts as the nanochannel. In Ω_4B_, there is only one chloridion and or sodion. Hence, water purification can be guaranteed when a slimmer CTT acts as a nanochannel (Appendix A).

In counting the numbers of water molecules in the three containers (Figure 7c, Table 3), we find Ω_5A_ and Ω_5C_ have more water molecules when a slimmer CTT is used. Since the piston moves at the same speed, the discharge of solution from Ω_1_ should be identical. Hence, Ω_4B_ must have less water molecules than Ω_5A_ or Ω_5C_. It implies that the water purification speed of the system is lower when a slimmer nanochannel is adopted. The purification rate via a thicker CTT is high (Table 3). Hence, a thicker CTT acting as the nanochannel will benefit water purification. Since thicker CTTs are easier to be fabricated, the present model is more feasible.

### 3.4. Effect of Solute Concentration on Ion Separation

What will happen to the ion separation process if the concentration of NaCl in the solution changes? To answer this question, we compare three cases with different concentrations of *C*_NaCl_ = 0, 0.6, and 1 mol/L. Figure 8a indicates that the three containers (Ω_5A_, Ω_4B_, and Ω_5C_) have more ions in the same duration when the solution has a higher concentration of NaCl. It is a drawback for the purification of water since more ions entering Ω_4B_ (Appendix A). According to the number difference of the positive and negative ions in each container, the electric charge of either Ω_5A_ or Ω_5C_ is independent of the concentration (Figure 8b). It is because the aqua ions in the containers occupy the entrance (or the exit of Ω_4A_/Ω_4C_), meanwhile, the aqua ions in the separating zone providing an inverse electric field that balances the external field. In this case, the cations and anions in the middle part of the separating zone are not well separated (comparing the snapshots at 1600 and 4850 ps in Figure 5b). Therefore, the ions may enter any of the three containers. Figure 8c gives the number of water molecules in the three containers, it implies that the separation quality of water is poorer when the solution has a higher concentration of solute. The issue can be dealt with by adopting larger containers for the aqua ions or enhancing the higher speed of flow in the separating zone.

When pure water (*C*_NaCl_ = 0) is in the resource reservoir, Ω_5A_ and Ω_5C_ have more water molecules than Ω_4B_, which is due to the effective cross section for mass transfer of Ω_3B_ is smaller than those of Ω_3A_ and Ω_3C_. However, when the concentration of solute is non-zero, the water molecule numbers in the three containers are slightly different. They are independent of the concentration of solute, as well. The reason is that the aqua ions in the electric field generate pressure on the entrance of the three subchannels (i.e., the left ends of Ω_3A_, Ω_3B_, and Ω_3C_), thus improve the effective cross section for mass transfer of Ω_3B_. This phenomenon contributes a clue to design a device for testing the purity of water. For example, when the amounts of water in the three containers are different obviously in the same duration (e.g., 1 ms), the water is thought pure. Otherwise, some ions exist in water when the three containers have slight difference of water amount.

## 4. Conclusions

We proposed a capacitive deionization model for ion separation and water purification. In the model, the CNT-based concentric twin tube (CTT) has three subchannels for separating the aqua ions and water, respectively. A capacitor with two electrodes parallel to the flow direction of solution generates electrostatic forces on the ions before they enter different containers via the subchannels. Using molecular dynamic simulation approach, we evaluated the physical quantities of the model, e.g., the variation of potential energy of the components, the numbers of ions and water molecules in different zones in the system, the electric charges of the containers with aqua ions, and the water purification rate. Following remarkable conclusions are drawn based on the results.

First, without an electric field, the ions are not separated before entering the containers. If the intensity of electric field is too high, e.g., the intensity >14 V/Å in this model, the inner tube of CTT has large deformation, which may result in failure of mass transfer. Under an appropriate electric field, both water purification and ion separation can be realized.

Secondly, by moving the piston at 0.01 Å/ps and using the A-30 CTT to separate the ions, the flux of purified water by the model can reach 20.41 kmol·m^−2^·s^−1^.

Thirdly, we observed the cavitation phenomenon when letting the solution flow from the source reservoir to the CTT channel. It is caused by the electric field-induced separation of the positive and negative aqua ions in the separating zone in CTT.

Fourthly, a thicker CTT acting as the nanochannel will benefit water purification during ion separation.

Finally, the ion separation effect depends on the solute concentration when the external pressure and electric field are specified. The reason is that the aqua ions may choke the subchannels when the concentration is high.

In the present study, the reaction between the aqua ions and the carbon frame is not considered. Moreover, the three containers have no gap at their boundaries that influences the power of the battery made from the containers. In our future work, the concerns will be discussed.

## Figures and Tables

**Figure 1 ijms-24-06677-f001:**
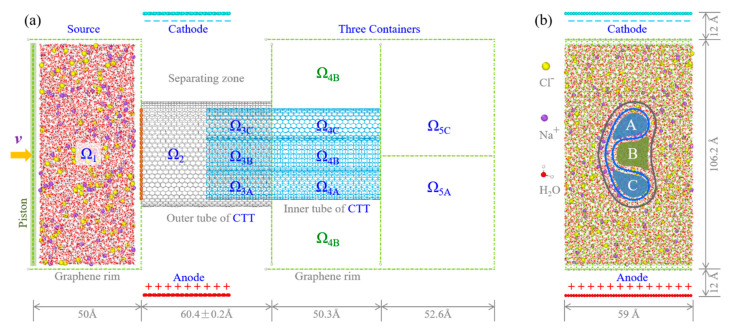
The unit cell of a membrane model for ion separation. (**a**) Side-view and (**b**) axial-view of the system. It has four essential components. First, the source reservoir Ω_1_ contains NaCl aqueous solution for separation. A piston at the left side of Ω_1_ provides an external pressure on the solution. It moves at a constant speed v = 0.01 Å/ps. Second, three containers (Ω_4B_, Ω_5A_, and Ω_5C_) are separated by graphene rims. Third, a concentric-twin-nanotube (CTT) channel connects the source reservoir and the three containers. The final component is the capacitor made from dual-charged graphene ribbons with size of 39 Å × 59 Å. It provides an electric field that covers part of the outer tube of CTT. The overlap of the two tubes in the CTT contains three subchannels, i.e., Ω_3A_, Ω_3B_, and Ω_3C_, for ion separation.

**Figure 2 ijms-24-06677-f002:**
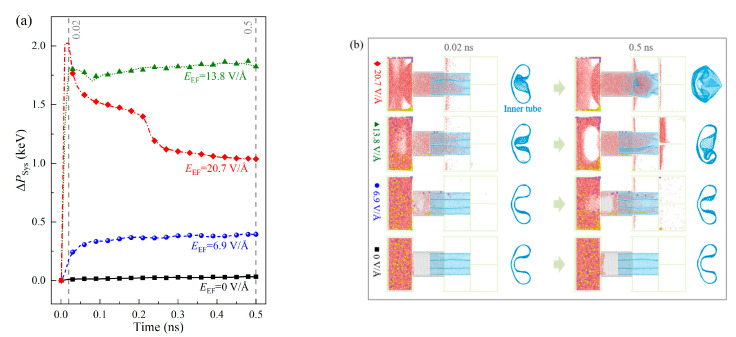
Variation of potential energy (VPE, Equation (1)) of the system under the electric field with different intensities (*E*_EF_) in Stage II. (**a**) VPE curves, (**b**) representative snapshots of the system with A-30 CTT as the channel. The initial concentration of NaCl in solution (*C*_NaCl_) in the source reservoir Ω_1_ is 0.6 mol/L.

**Figure 3 ijms-24-06677-f003:**
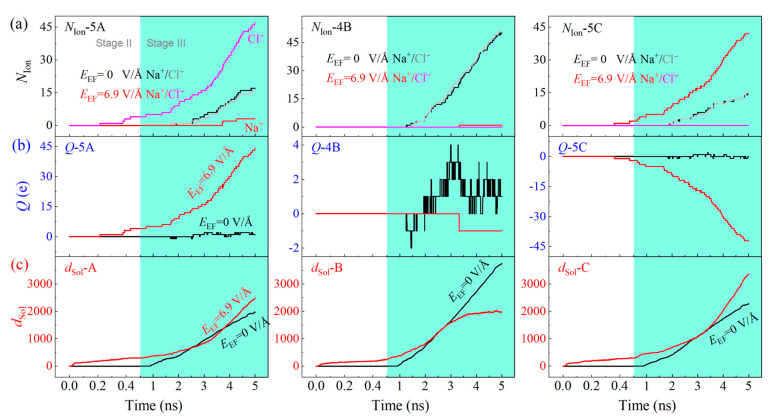
Physical quantities of the solution with *C*_NaCl_ = 0.6 mol/L flowing through the A-30 CTT nanochannel in the system with or without the electric field with the intensity (*E*_EF_) of 6.9 V/Å. (**a**) The number of ions in the three containers, (**b**) the electric charges in the three containers, and (**c**) the discharge of solution via the three subchannels in each CTT.

**Figure 4 ijms-24-06677-f004:**
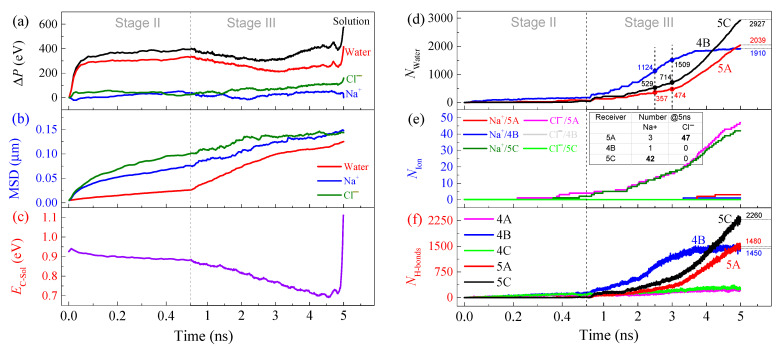
Physical quantities of the ion separation model with the A-30 CTT as subchannels at 300 K when *E*_EF_ = 6.9 V/Å and *C*_NaCl_ =0.6 mol/L NaCl. (**a**) VPE of the components in solution. (**b**) Mean square displacements (MSD) of the components. (**c**) Interaction energy between solution and the carbon frame. (**d**) The numbers of the water molecules in the three containers. (**e**) The numbers of ions in the three different subchannels and their connected containers. (**f**) The numbers of H-bonds in the three subchannels and their connected containers.

**Figure 5 ijms-24-06677-f005:**
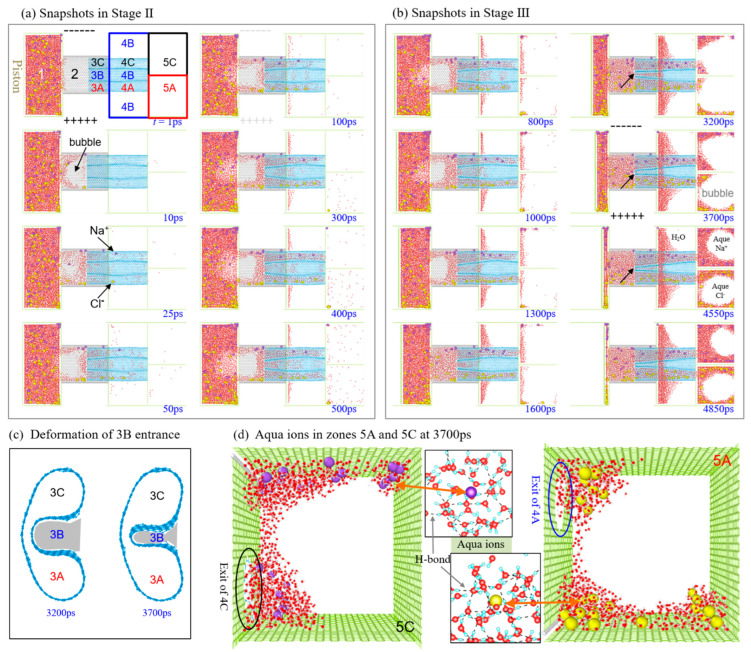
Motion of the solution via the A-30 CTT as subchannels at 300 K when *E*_EF_ = 6.9 V/Å and *C*_NaCl_ = 0.6 mol/L. (**a**) Snapshots in Stage II (just open the valve), (**b**) snapshots in Stage III (push the piston at the speed of *v*). The black arrows indicate the entrance of Ω_3B_. (**c**) Choked entrance of Ω_3B_ due to large deformation. (**d**) Aqua ion distribution in Ω_5A_ and Ω_5C_ at 3700 ps.

**Figure 6 ijms-24-06677-f006:**
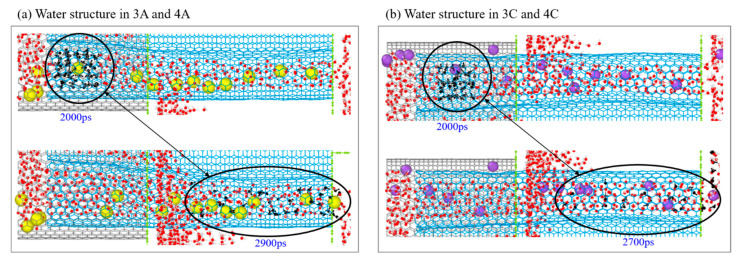
Deformation of water structure in the A-channel and C-channel. (**a**) In Ω_3A_ and Ω_4A_; (**b**) In Ω_3C_ and Ω_4C_ zones. The initial water cluster (black molecules) at the left end of Ω_3A/3C_ disperses before exit from Ω_4A_ or Ω_4C_.

**Figure 7 ijms-24-06677-f007:**
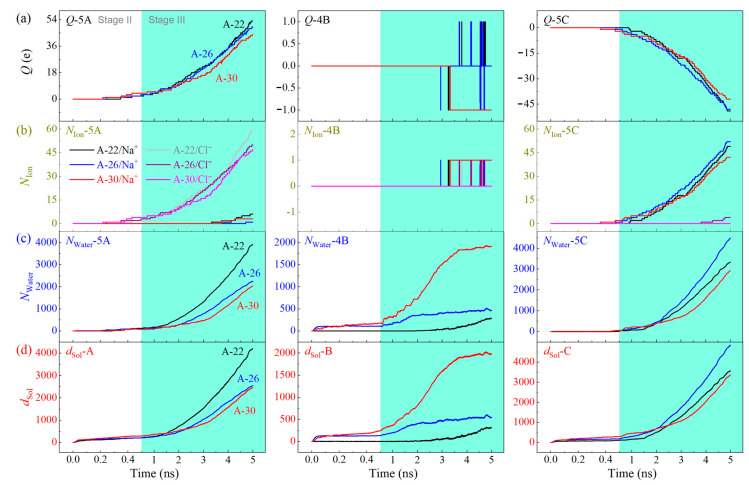
Physical quantities of the solution with *C*_NaCl_ = 0.6 mol/L flowing through the three different CTT nanochannels when *E*_EF_ = 6.9 V/Å. (**a**) Electric charges in the three containers, (**b**,**c**) the numbers of ions and water in the three containers, and (**d**) the discharge of solution via the three subchannels in each CTT.

**Figure 8 ijms-24-06677-f008:**
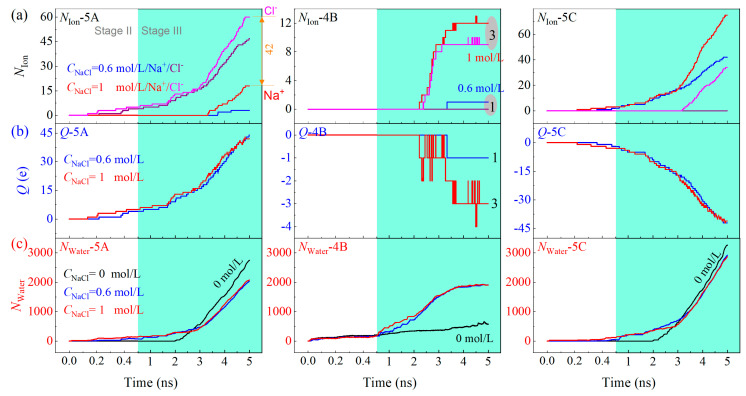
Physical quantities of the solution with different concentration flowing through the A-30 CTT nanochannel when *E*_EF_ = 6.9 V/Å. (**a**) The number of ions in different containers, (**b**) the electric charges in the three containers, and (**c**) the number of water molecules via the three subchannels.

**Table 1 ijms-24-06677-t001:** Number of water molecules and NaCl in the solution with different concentrations.

Concentration	NaCl	H_2_O
0 mol/L	0	9330
0.6 mol/L	103	9250
1 mol/L	176	9180

**Table 2 ijms-24-06677-t002:** Parameters in the potential functions for describing the nonbonding interactions.

Element	*ε* (kcal/mol)	*σ* (Å)	*q* (e)
C [31]	0.0860	3.4000	0.0000
O [48]	0.1553	3.1660	−0.8476
H (H_2_O) [48]	0.0000	0.0000	0.4238
H (C-H)	0.0000	0.0000	0.0000
Na^+^ [52]	0.3526	2.1595	1.0000
Cl^−^ [52]	0.0128	4.8305	−1.0000

**Table 3 ijms-24-06677-t003:** Physical quantities of the solution separated by the three subchannels at 5 ns.

Zone	*Q* (e)	*N* _Water_	*d* _Sol_
A-22	A-26	A-30	A-22	A-26	A-30	A-22	A-26	A-30
A/5A	−53	−49	−44	3916	2251	2039	4200	2546	2474
B/4B	0	0	1	288	465	1910	313	544	1973
C/5C	49	48	42	3331	4481	2927	3569	4844	3359
*R* _Puri_				3.82%	6.46%	27.78%

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
