# Peer review of "Ion Separation Together with Water Purification via a New Type of Nanotube: A Molecular Dynamics Study"

_ijms, 2023, doi:10.3390/ijms24076677_

Round 1

Reviewer 2 Report

The present manuscript proposes a CNT-based concentric twin tube (CTT) nanochannel for both water purification and ion separation. Both the study and the results are very interesting.  On the other hand, the results are not discussed with relevant research articles. Therefore, the manuscript is written as a report. In addition, a validation of the numerical model that was used is missing from the manuscript. Therefore, the manuscript should be reconsidered for publication in International Journal of Molecular Sciences after major revision. The following comments should be taken into account.

1. The originality of the paper needs to be stated clearly. It is of importance to have sufficient results to justify the novelty of a high-quality journal paper. The Introduction should make a compelling case why the study is useful along with a clear statement of its novelty or originality by providing relevant information and providing answers to basic questions such as: What is already known in the open literature? What is missing (i.e., research gaps)? What needs to be done, why and how? Clear statements of the novelty of the work should also appear briefly in the Abstract and Conclusions sections.

2. Authors should add more explanation about this investigation’s contribution in the introduction section.  

3. The results are not discussed in the manuscript. Authors should discuss the results and how they can be interpreted in perspective of previous studies and of the working hypotheses. The findings and their implications should be discussed in the broadest context possible and limitations of the work highlighted. Please include and discuss at least ten relevant studies.

4. A validation of the numerical model is needed.

5. The literature review should be updated with other methods for the purification of water. The following articles may be used:

a)         Micromixing efficiency of particles in heavy metal removal processes under various inlet conditions, Water (Switzerland), 2019, 11(6), 1135.

b)         A computational tool for the estimation of the optimum gradient magnetic field for the magnetic driving of the spherical particles in the process of cleaning water, Desalination and Water Treatment, 2017, 99, 27–33.

c)         Numerical study of magnetic particles mixing in waste water under an external magnetic field, Journal of Water Supply: Research and Technology - AQUA, 2020, 69(3), 266–275.

d)         Mixing of Fe3O4 nanoparticles under electromagnetic and shear conditions for wastewater treatment applications, Journal of Water Supply: Research and Technology-Aqua, 2022, 71 (6), 671–681.

Round 2

Reviewer 2 Report

The manuscript can now be accepted for publication